# Integration of Porous Nanomaterial-Infused Membrane in UF/FO Membrane Hybrid for Simulated Osmosis Membrane Bioreactor (OsMBR) Process

**DOI:** 10.3390/membranes13060577

**Published:** 2023-06-01

**Authors:** Ahmadreza Zahedipoor, Mehdi Faramarzi, Amir Mansourizadeh, Abdolmohammad Ghaedi, Daryoush Emadzadeh

**Affiliations:** 1Department of Chemical Engineering, Membrane Science and Technology Research Center (MSTRC), Gachsaran Branch, Islamic Azad University, Gachsaran P.O. Box 75818-63876, Iran; 2Department of Chemistry, Gachsaran Branch, Islamic Azad University, Gachsaran P.O. Box 75818-63876, Iran; 3Department of Chemical and Biological Engineering, University of Ottawa, Ottawa, ON K1N 6N5, Canada

**Keywords:** porous titanium dioxide, ultrafiltration, nanocomposite membrane, osmosis membrane bioreactor, wastewater treatment, antifouling

## Abstract

This study explored the use of a combination of hydrothermal and sol–gel methods to produce porous titanium dioxide (PTi) powder with a high specific surface area of 112.84 m^2^/g. The PTi powder was utilized as a filler in the fabrication of ultrafiltration nanocomposite membranes using polysulfone (PSf) as the polymer. The synthesized nanoparticles and membranes were analyzed using various techniques, including BET, TEM, XRD, AFM, FESEM, FTIR, and contact angle measurements. The membrane’s performance and antifouling properties were also assessed using bovine serum albumin (BSA) as a simulated wastewater feed solution. Furthermore, the ultrafiltration membranes were tested in the forward osmosis (FO) system using a 0.6-weight-percent solution of poly (sodium 4-styrene sulfonate) as the osmosis solution to evaluate the osmosis membrane bioreactor (OsMBR) process. The results revealed that the incorporation of PTi nanoparticles into the polymer matrix enhanced the hydrophilicity and surface energy of the membrane, resulting in better performance. The optimized membrane containing 1% PTi displayed a water flux of 31.5 L/m^2^h, compared to the neat membrane water value of 13.7 L/m^2^h. The membrane also demonstrated excellent antifouling properties, with a flux recovery of 96%. These results highlight the potential of the PTi-infused membrane as a simulated osmosis membrane bioreactor (OsMBR) for wastewater treatment applications.

## 1. Introduction

Wastewater treatment is a critical issue for public health and the environment. Industrial and domestic wastewater contains contaminants that can cause serious harm to human health and aquatic life if not treated effectively [1]. Membrane-based filtration systems are widely used for wastewater treatment because of their high efficiency, low energy consumption, and ease of operation [2]. 

An osmosis membrane bioreactor (OsMBR) is an innovative technology that combines osmosis and bioreactor processes for wastewater treatment [3]. The OsMBR system utilizes a semi-permeable membrane to separate the wastewater into two flows, a concentrated flow and a permeating flow. The concentrated flow contains high-concentration wastewater, while the permeated flow contains purified water. The osmosis process occurs through a pressure gradient that causes water molecules to move from the high-concentration flow to the low-concentration flow [4]. In the bioreactor flow, microorganisms are used to break down organic matter and remove pollutants from the wastewater. The treated wastewater is then circulated back into the concentration flow, creating a continuous cycle. The use of a membrane in the OsMBR system provides several advantages, including a high degree of separation, reduced footprint, and lower energy consumption compared to conventional wastewater treatment methods [5]. However, membrane fouling remains a significant challenge that limits its performance and lifespan [6,7]. Therefore, researchers have been exploring innovative ways to improve membrane performance and reduce fouling [8,9,10].

Recent studies have focused on enhancing the performance of the OsMBR system by improving the membrane properties, such as hydrophilicity and permeability, to reduce fouling and enhance the system’s lifespan [11]. The use of nanoparticles in the OsMBR process has been explored to enhance its performance [12]. Nanoparticles can improve the permeability of the membrane, increase the removal efficiency of organic and inorganic pollutants, and prevent membrane fouling. In this context, various types of nanoparticles, including silver [13], titanium dioxide [14], graphene oxide [5], and carbon nanotubes [15], have been investigated for their potential application in MBRs. The literature suggests that the incorporation of nanoparticles in MBRs has shown promising results for improving the treatment efficiency and reducing the operational costs of the process [15]. 

On the other hand, porous nanoparticles offer several advantages over nonporous nanoparticles for a variety of applications. Firstly, their high surface-area-to-volume ratio provides a larger surface area for a possible reaction. Secondly, their porous structure allows for the easy diffusion of small molecules or ions, which can improve their performance as adsorbents or separation agents. Additionally, the pore size and shape of the nanoparticles can be tailored to specific applications, allowing for selectivity and specificity. The porous structure also provides mechanical stability and can help prevent the aggregation or degradation of the nanoparticles. Lastly, the porous nature of the nanoparticles can be used to encapsulate or immobilize other materials, such as drugs or enzymes, for targeted delivery or catalysis [16,17].

More importantly, porous titanium dioxide (TiO_2_) is a type of nanoparticle that has gained significant attention due to its unique properties and potential applications. The porous structure of TiO_2_ nanoparticles allows for improved surface area and reactivity, making them effective for various chemical reactions [18]. The photocatalytic properties of TiO_2_ nanoparticles arise from their ability to absorb light energy and create electron–hole pairs, which can then react with molecules to generate reactive oxygen species or reduce pollutants. Porous TiO_2_ nanoparticles can also be used as adsorbents for removing contaminants from water or air due to their high surface area and adsorption capacity [19]. Additionally, porous TiO_2_ nanoparticles can be used as a drug delivery system [20] or in biomedical applications, as their porous structure can be tailored for controlled drug release or cellular uptake. As a result, porous TiO_2_ nanoparticles have shown great potential in a variety of fields, including environmental remediation, catalysis [21], and biomedical applications [22]. In water treatment applications, compared to nonporous TiO_2_ nanoparticles, porous TiO_2_ nanoparticles demonstrate a greater surface area, which enhances their capacity to adsorb organic and inorganic contaminants during water treatment. This, in turn, can augment the removal efficiency and performance of the membrane. Moreover, the porous nature of TiO_2_ nanoparticles can enhance the hydrophilicity of the membrane surface, leading to improved water transport and reduced fouling by preventing the adsorption of organic and inorganic substances. The porous structure also promotes the stability and durability of the membrane, making it more resistant to damage and deformation [23].

This study aims to evaluate an osmosis membrane bioreactor using a hybrid approach that combines an ultrafiltration (UF) membrane and the forward osmosis (FO) process. The FO process involves separating a solution with a lower concentration of dissolved solids (the feed solution) from a solution with a higher concentration (the draw solution) through a semi-permeable membrane. Meanwhile, the UF membrane in the hybrid system is used to filter out large molecules, such as BSA, and investigate the fouling effect, which is a major issue in membrane bioreactors. By using this hybrid approach, the FO process can effectively separate water molecules from dissolved solids while minimizing the fouling effect. Furthermore, this study aims to contribute to the development of advanced membrane materials and technology for wastewater treatment. The results of this research provide insights into the potential of PTi-infused membranes as an efficient and antifouling membrane bioreactor for wastewater treatment applications. Overall, this study demonstrates the importance of innovative approaches to address the challenges of wastewater treatment and the potential of porous nanomaterials to enhance membrane performance and reduce fouling.

## 2. Materials and Experimental

Polyvinyl pyrrolidone (PVP K30, Sigma Aldrich, Darmstadt, Germany) with a molecular weight of 5500 g/mol and polysulfone Udel P-1700 (Solvay Advanced Polymers, Düsseldorf, Germany) were applied for polymeric membrane fabrication. N-Methyl-2-pyrrolidone (NMP), a solvent from the Sigma Aldrich Company, was used. To create the PTis, ethanol, Ti(SO_4_)_2_, and cetyltrimethylammonium bromide (CTAB, Sigma Aldrich) were purchased from Sigma Aldrich. To prepare the simulated wastewater, bovine serum albumin (BSA) (Sigma Aldrich) with a molecular cutoff of 66 KDa was employed. To provide sufficient driving forces, poly (sodium 4-styrene-sulfonate, 70 kDa, Sigma Aldrich) was utilized as a draw solution.

### 2.1. Synthesis of Porous Titanium Dioxide 

Initially, Ti(SO_4_)_2_ was dissolved in distilled water, and a cetyltrimethylammonium bromide (CTAB) solution was added to the mixture. The resulting mixture was then stirred for 30 min before being allowed to settle for 12 h at room temperature. Subsequently, the mixture was transferred to an autoclave and subjected to hydrothermal treatment at 100 °C for 72 h. Once complete, the powders were separated from the solution through centrifugation and washed with water and ethanol to remove any organic impurities. Finally, the samples were calcined at 165 °C temperature for 6 h to promote crystallization [24].

### 2.2. Preparation of Nanocomposite Membrane

Four distinct types of membranes were produced by using fixed values of PSF, PVP, and NMP and different concentrations of nanoparticles. Prior to the addition of the necessary quantity of PTi to the solution, 0.5 PVP was added to the NMP solvent at the prescribed concentration. The solution was then vigorously mechanically stirred for 24 h at 25 °C before being allowed to degas for 4 h. Finally, the solution was cast using an adjustable blade to construct the membrane. The resulting membrane was then immersed in distilled water and maintained in a water bath at room temperature until phase inversion occurred. Based on the nanoparticle concentration, the membranes were assigned names (MT0 control, MT0.5, MT1, and MT2) as listed in Table 1 [25].

### 2.3. UF/FO Membrane Hybrid for Simulated Osmosis Membrane Bioreactor (OsMBR) Process 

One of the main objectives of this study was to explore the use of an ultrafiltration (UF) membrane in a forward osmosis process (hybrid UF/FO) as a simulated osmotic membrane bioreactor (OsMBR) application. In this work, BSA was used as a foulant model to simulate organic wastewater, which is a major issue in MBRs. To generate the driving force for the FO process, poly (sodium 4-styrene-sulfonate, 70 kDa) was used at a concentration of 0.6% as the draw solution, resulting in a pressure of 0.6 bar [26] on the UF membrane surface. Poly (sodium 4-styrene-sulfonate, 70 kDa) was used since it would not pass through the UF membrane (125 KDa) [27] due to its higher molecular weight cutoff. By utilizing UF and FO hybrid membrane in this manner, this study aimed to enhance the performance and efficiency of MBRs in treating organic wastewater. 

Figure 1 provides a simplified representation of the forward osmosis system. The membrane cell was designed with a cross flow, and the membrane had an effective area of 14 cm^2^ within the cell. Two pumps were utilized to cycle the feed and draw solutions through the system while maintaining a constant temperature of 25 °C. In order to measure the water flow, two digital balances were positioned beneath each tank [28]. This setup consisted of three main components: a feed tank, a draw tank, and a membrane module. The feed and draw solutions were typically stored in their respective tanks and circulated through the membrane module by pumps. 

The water flux $j_{v}$ (L/m^2^ h) was calculated by the following equation:(1)$$j_{v}=\frac{\Delta V}{A_{m}.\Delta t}=\frac{\frac{\Delta m}{\rho }}{A_{m}.\Delta t}$$

In this equation, $A_{m}$ is the effective surface of the membrane (in square meters), Δt is the penetration time (in hours), Δm is the change in the weight of the osmosis solution, and ρ is the density of the feed. Osmosis membrane bioreactor tests were conducted at 25 °C temperature.

The solute flux $j_{s}$ (L/m^2^ h) was obtained by utilizing the following equation [28]:(2)$$J_{s}=\frac{\Delta \left(C_{t}.V_{t}\right)}{A_{m}.\Delta t}$$
where, in this equation, ΔV is the feed volume change, and ΔC is the concentration change at 25 °C [22].

Bovine serum albumin (BSA) was used for the preparation of the simulated wastewater (feed solution), and 0.6% poly (sodium 4-styrene-sulfonate), which is equal to 0.6 bar, was applied as an osmosis solution because of its particle size, which can be recovered by UF [29]. 

### 2.4. Membrane Fouling Tests

Fouling experiments were performed in a lab setting employing an osmosis membrane bioreactor system and a crossflow filtration unit. The input flow was changed from pure water to the effluent solution (BSA) after 60 min of pure water filtration. Then, the measurement of the solution that penetrated through the membrane (j_p_ (kg/m^2^h)) was started after 10 min of the flow change. The membrane was cleaned with flowing distilled water without applying any pressure for 30 min after filtering the effluent solution for 210 min. After that, the test for clean water filtration continued for an additional 60 min. Therefore, 370 min was determined to be the overall filtering time. The flux recovery ratio (FRR) is given as [30]:(3)$$\mathrm{FRR}=\left(\frac{j_{w2}}{j_{w1}}\right)$$

Thus, $j_{W1}$ is the net flow of water at 60 min of filtration, and $j_{W2}$ is the net flow of water at 285 min of filtration. Generally, a higher FRR shows better antifouling properties in membranes. Furthermore, to analyze the fouling process in detail, the total fouling rate ($R_{t}$), reversible fouling rate ($R_{r}$), and irreversible fouling rate ($R_{\mathrm{ir}}$) were computed by applying the following equations [30].
(4)$$R_{t}\left(\%\right)=\left(1-\frac{j_{p}}{j_{w1}}\right).$$
(5)$$R_{r}\left(\%\right)=\left(\frac{j_{w2}-j_{p}}{j_{w1}}\right)\times 100$$
(6)$$R_{\mathrm{ir}}\left(\%\right)=\left(\frac{j_{w1}-j_{w2}}{j_{w1}}\right)\times 100=R_{t}-R_{r}$$

In this regard, $j_{p}$ is the permeate flow during 210 min of filtration with the simulated sludge solution feed (BSA).

### 2.5. Membrane Characterization 

The properties of the inorganic particles and membranes were characterized using a variety of techniques, including transmission electron microscopy (TEM), scanning electron microscopy (SEM, TM3000, Hitachi, Tokyo, Japan), Fourier transform infrared spectroscopy (FTIR; FTLA 2000 series, ABB, Tokyo, Japan), surface area analysis, atomic force microscopy, X-ray diffraction (D/max-rB 12 kW Rigaku, Tokyo, Japan), and contact angle measurements. Finally, the contact angle of the membranes was measured using the sessile drop method using a contact angle goniometer (IMC-159D, IMOTO Machinery, Kyoto, Japan). Brunauer–Emmett–Teller (BET) and Barrett–Joyner–Halenda (BJH) tests were conducted to determine the surface area and the pore size distribution of the nanoporous TiO_2_ nanoparticles.

The morphology of the resulting nanocomposite membranes was analyzed using atomic force microscopy (AFM, Park NX10). AFM images were acquired in tapping mode, scanning over a 10 × 10 μm^2^ area with a 512 × 512-pixel-width resolution and an adaptive scan rate ranging from 0.2 to 2 Hz. The surface topography of the membranes was evaluated in terms of several parameters, including the root mean square of height deviations (Rq), the average plane roughness (Ra), the maximum peak-to-valley distance (Rz), and the ratio of the actual surface area for a rough surface to the planar area (r). A value of r=1 indicates a perfectly smooth surface. Two samples of each membrane were analyzed using AFM. 

To characterize the intrinsic surface hydrophilicity of the membranes, the solid–liquid interfacial free energy (Δ*G_SL_*) was determined, since the apparent contact angle can be influenced by surface roughness [31]. The $\Delta G_{SL}$ value (in mJ/m^2^) was calculated using the Young–Dupre equation:(7)$$\Delta G_{SL}=\gamma_{l}\left(1+\frac{\cos\theta }{r}\right)$$
where *θ* is the average water contact angle, which decreases as surface hydrophilicity increases (i.e., cos*θ* increases), and the water surface tension ($\gamma_{l}$) has a value of 72.8 mJ/m^2^ at 25 °C. The roughness area ratio (r), obtained from the AFM topographical images, is also included in the equation. The minimum value of r for an ideally smooth surface is unity, which maximizes $\Delta G_{SL}$ for a given hydrophilicity. As the hydrophilicity of the surface increases, the value of $\Delta G_{SL}$ increases as well [31]. 

## 3. Results and Discussion

### 3.1. Characterization of the Nanoparticles and Membrane

Figure 2 presents a TEM image of titanium dioxide nanoparticles, with sizes ranging from 100 to 200 nm. An analysis of the image (including 85 particles) indicated an average particle size of 18.99 nm, with the smallest particle at 7.64 nm and the largest at 41.48 nm. The varied shapes of the particles suggested the crystal structure of the samples. Additionally, the presence of empty spaces between the particles indicated the existence of pores in the sample, albeit in small quantities. This porosity can potentially enhance the surface area of the nanoparticles, which can enhance water transport and improve its efficiency [32].

The two main methods for determining nanoporous nanoparticles’ pore size distribution and surface area parameters are the BET (Brunauer–Emmett–Teller) and BJH (Barrett–Joyner–Halenda) methods. The BET method is used to calculate the specific surface area of a material, which is a measure of the total surface area of the pores per unit mass of the material. The BJH method, on the other hand, is used to determine the pore size distribution of a material [33].

The results of the BET and BJH experiments for the determination of the surface area and pore size distribution of nonporous TiO_2_ are shown in Figure 3. The nitrogen adsorption–desorption isotherm curve of the nanoparticles exhibits an H3 hysteresis loop and a range of 0.66 to 0.99 P/Po, classified as a type-IV isotherm. The results indicate that the porous TiO_2_ nanoparticles have a specific surface area of 112.84 m^2^/g, as measured by BET (Figure 3a). The total pore volume was found to be 0.2186 cc/g at P/P0 = 0.981, according to Figure 3b. According to Figure 3c, the pore size distribution ranges are between 2 and 75 nm, which indicates a mesoporous structure for the synthesized nanoparticles. The high BET surface area and mesoporous structure of the nanoparticles have the potential to enhance the membrane performance.

Figure 4 illustrates the X-ray diffraction (XRD) pattern of titanium dioxide nanoparticles (PTis), the neat membrane, and the membrane modified with titanium dioxide nanoparticles. The XRD pattern of PTis revealed the presence of the anatase crystal phase, with distinct diffraction angles at 25°, 38°, 47°, 55°, and 63°. The XRD pattern of the pure membrane showed diffraction angles at 25, 31, 35, 41, and 45 degrees. The XRD pattern of the membrane modified with titanium dioxide nanoparticles and polysulfone showed diffraction angles at 25° (weaker), 30°, 35°, 41°, 45°, and 51°. The XRD pattern of the nanocomposite membrane showed the presence of the anatase crystal phase at an angle of 25°, albeit with reduced peak intensity, suggesting an interaction between the titanium dioxide nanoparticles and the membrane surface. This interaction indicates that the surface of the nanocomposite membrane has been modified [34]. 

The morphologies of the membranes were analyzed using scanning electron microscopy (SEM). The SEM images of the pristine and nanocomposite membranes are shown in Figure 5. It can be observed that as the concentration of nanoparticles increases, the size of the pores on the membrane surface gradually decreases. This change in morphology can have an impact on both water flux and BSA rejection. The incorporation of nanoparticles during the phase inversion process may be responsible for this alteration, as it can increase the hydrophilicity of the polymer solution. The increased hydrophilicity of the dope solution can facilitate water diffusion while maintaining the outflow rate of NMP. This can lead to the creation of more pores with smaller sizes, resulting in an enhanced porous structure [35]. As a result, the asymmetric membrane constructed with more nanoparticles has a higher porosity, as the higher hydrophilicity improves the flow of non-solvent (water) in the solvent/non-solvent exchange. The concentration of nanoparticles in the membrane directly affects its hydrophilic characteristics, as demonstrated by the decrease in the contact angle and porosity, which are both enhanced with increasing nanoparticle concentration [36].

Figure 6 shows the ATR-FTIR spectra of the PSf (control), and nanocomposite membranes. In the PSf (control), the peaks at specific wave numbers of 1150 cm^−1^ (symmetric O-S-O stretching), 1300 cm^−1^ (asymmetric O-S-O stretching), 1250 cm^−1^ (asymmetric C-O-C stretching), 1500 cm^−1^ (CH_3_ bending vibration), and 1405 cm^−1^ (C-C aromatic ring stretching) correspond to the specific functional groups of the PSf polymer substrate [36]. The FTIR spectra of the nanocomposite membrane showed reduced peak C=O (carbonyl) at around 1680 cm^−1^ compared to the unmodified membrane, indicating the presence of titanium dioxide nanoparticles and their possible hydrogen interaction with functional OH groups of the PTis and active site of nanocomposite membrane.

The smoothness and roughness of the membrane surfaces were analyzed using atomic force microscopy (AFM). Figure 7 shows four 3D AFM images of polysulfone membranes, including the unmodified membrane and membranes modified with porous titanium dioxide nanoparticles. The unmodified polysulfone membranes (Figure 7a) exhibited a lower degree of surface unevenness and roughness than the membranes modified with titanium dioxide nanoparticles (Figure 7b–d). This indicates that the unmodified membrane may have better antifouling properties due to its higher smoothness, lower unevenness, and uniform color. Table 2 summarizes the roughness of the plain and modified membranes, including the average roughness (Ra), the root mean square of data (Rz), the height difference between the highest peak and the lowest valley (Rms), the roughness area ratio (R), and surface energy ($\Delta G_{SL}$). Surface roughness also affects deposition characteristics, with rougher surfaces increasing the likelihood of pollutants adhering and being trapped, leading to a higher deposition potential. These results are consistent with Figure 7, which demonstrates a significant increase in Ra from 20.9 nm for the bare membrane to 42.6 nm for the MT2-modified membrane. However, the parameter r (roughness area ratio), which affects $\Delta G_{SL}$, did not follow the same trend as Rq, Ra, and Rz. When the nanoparticle loading increased, Rq, Ra, and Rz increased, but r decreased. It is worth noting that large values of Rq, Ra, and Rz, with corresponding small values of r, indicate less aggregation of foulants on the membrane surface [37]. In fact, higher surface energy ($\Delta G_{SL}$) tends to lower fouling. As a result, an increase in the number of nanoparticles can increase the surface energy from 93.9 to 107.5, which indicates a lower tendency toward fouling.

In addition to this, the contact angle, determined using the sticky drop approach, is also shown in Table 2. As the concentration of PTis in the membranes increased, the contact angle decreased, indicating improved hydrophilic characteristics of the membrane surface. This is likely due to the presence of functional groups in PTis, which reduce the contact angle and increase the surface energy, resulting in lower deposition fouling potential on the membrane [38]. 

The elemental composition of the pure polysulfone membrane and the modified polysulfone membrane with added porous titanium dioxide nanoparticles was analyzed using X-ray photoelectron spectroscopy (XPS), and the results are presented in Table 3. The XPS spectra of the modified membranes revealed three peaks corresponding to the binding energies of carbon (C), oxygen (O), and nitrogen (N) atoms. The analysis of the spectra of the modified membranes, made from a combination of polysulfone and porous titanium dioxide nanoparticles, showed different percentages of these elements. This confirmed the presence of nanoparticle compounds in the modified membranes that are not present in the pure polysulfone membrane. These findings further support the conclusion that the modified membranes have improved properties due to the incorporation of titanium dioxide nanoparticles [31].

### 3.2. Membrane Performance

This section reports on the water and reverse solute flux through PSf and nanocomposite membranes during the UF/FO hybrid process as the simulated OsMBR. The addition of porous titanium dioxide nanoparticles to the nanocomposite membrane resulted in significantly higher fluxes compared to the PSf membrane. Figure 8 illustrates that increasing the concentration of nanoparticles in the nanocomposite membrane improved water flux from 13.7 to 38.8 L/m^2^h for 2% TPs, with the highest flux observed in the membrane containing 2% nanoparticles. These findings suggest that modifying the nanocomposite membrane with nanoparticles can enhance OsMBR performance. In fact, the use of these nanoparticles increases the surface-area-to-volume ratio of the membrane, which creates more active sites for water adsorption and results in improved water flux. Moreover, the use of nanoparticles with enhanced hydrophilicity can modify the membrane’s surface properties and increase its water flux [39]. 

The UF membrane, made of 18 wt% concentration of PSF polymer, is specifically engineered to allow only molecules with a molecular weight less than or equal to 125 kDa to pass through while retaining larger molecules. This selective filtration process is achieved by designing the pore size of the membrane in a way that separates molecules based on their size. Remarkably, the PSF UF membrane with a 125 kDa molecular cutoff is also able to separate molecules that are smaller than its molecular cutoff. For example, the membrane effectively retains molecules with a molecular weight of 66 kDa or 70 kDa, such as BSA in the feed solution and poly (sodium 4-styrene-sulfonate) in the draw solution, respectively. This selectivity is possible because of the size and structure of the membrane’s pores, which are specifically designed to be highly selective for certain molecular weights. With an increase in the concentration of nanoparticles in the membrane, the surface pores (as shown in SEM 5) decreased in size [40], leading to a reduction in the reverse solute flux from 2 to 0.5 g/m^2^h, as demonstrated in Figure 8. However, the increase in reverse solute flux observed in the MT2 membrane may be attributed to membrane defects caused by nanoparticle agglomeration, which may decrease the membrane’s efficiency due to limited compatibility between the nanoparticles and polymer matrix. These improvements were demonstrated through SEM images that show a decrease in pore size with an increase in nanoparticle concentration. The observed results suggest that the membrane containing 1 wt% nanoparticles had the best overall performance. This was evidenced by its higher water flux and lower solute flux compared to the other membranes [41].

This section examines the impact of incorporating porous TiO_2_ nanoparticles into a membrane on the membrane’s antifouling properties. Typically, high-quality membranes display high flux, a low fouling propensity, and consistent selectivity over time [42]. The behavior of the membrane surface plays a critical role in the fouling process, with hydrophobic surfaces typically being more prone to fouling by organic contaminants. To improve membrane permeability and antifouling properties, several methods have been proposed, including material modification, polymer composition, and surface modification [3]. Incorporating hydrophilic additives, combined with other methods, is one of the most effective ways to increase antifouling properties. To assess the effectiveness of composite membranes in preventing fouling, we tested a solution of BSA with a concentration of 400 ppm on four different membranes. The normalized water flux results for both the neat and nanocomposite membranes under a foulant feed are presented in Figure 9. The data show that as the concentration of nanoparticles in the nanocomposite membranes increases (MT0.5 and MT2), the reduction in water flux is significantly decreased compared to the neat membrane. For instance, the neat membrane exhibits a flux reduction of 35%, while the addition of 2% nanoparticles to MT2 leads to only a 5% reduction in water flux. These findings suggest that nanocomposite membranes with a higher nanoparticle concentration exhibit a lower tendency for fouling. In fact, the presence of nanoparticles alters the surface chemistry and morphology of the membrane, creating a hydrophilic and rougher surface. This roughness, combined with the presence of nanoparticles, inhibits the attachment of foulants, such as proteins or bacteria, to the membrane surface [40]. Additionally, the hydrophilic surface of the nanocomposite membrane promotes easy water flow and prevents the accumulation of organic matter on the surface, further reducing the fouling tendency [43]. Therefore, the use of nanocomposite membranes holds tremendous potential in reducing fouling and enhancing the efficiency of filtration processes in various applications.

Regarding the type of fouling, membrane fouling can be classified as reversible or irreversible, where reversible fouling occurs when fouling materials can be easily removed by washing with water, whereas irreversible fouling occurs when fouling materials cannot be removed by washing [44]. Backwashing can remove reversible fouling, but it reduces the membrane’s effectiveness and increases operational costs. Irreversible fouling must be removed through chemical cleaning, which shortens the membrane’s lifespan. Figure 10 presents the total fouling ratio (Rt), reversible hydraulic fouling ratio (Rr), and irreversible hydraulic fouling ratio (Rir) of the nanocomposite membranes. The results indicate that the modified membranes have lower resistance coefficients and greater flux recovery ratios. Rr and Rir together make up the overall deposition resistance of UF membranes manufactured with inserted PTis, as illustrated in Figure 10. The composite membranes with 0.5%, 1%, and 2% by weight of PTis had significantly lower irreversible resistance (Rir), decreasing from 44% for the unmodified membrane to 20%, 4%, and 5%, respectively, for the modified membranes. These findings demonstrate the strong antifouling capabilities of nanocomposite membranes. PTis create a more hydrophilic surface and higher membrane solid–liquid interfacial free energy ($\Delta G_{SL}$), which inhibits the attachment of foulants, such as proteins or bacteria, to the membrane surface [45]. 

The nanoparticles can also modify the surface chemistry and morphology of the membrane, creating a more hydrophilic surface, as reported in Table 3, which promotes easy water flow and prevents the accumulation of organic matter on the surface, further reducing the fouling tendency. Moreover, the incorporation of nanoparticles increases the surface area of the membrane, providing more active sites for water adsorption and improving water flux. The higher flux of water through the membrane reduces the residence time of foulants on the membrane surface, decreasing their attachment and penetration and thereby reducing Rir to less than 5%. In addition, the incorporation of nanoparticles also enhances the mechanical stability of the membrane, which reduces Rir. According to the literature [46], nanoparticles may improve the membrane’s mechanical properties by increasing its strength and stability, allowing it to withstand harsh conditions and prolong its lifespan. The increased mechanical stability also improves the membrane’s stability during cleaning procedures, reducing the risk of damage during cleaning [46].

The FRRs (flux recovery ratios) of the membrane samples are presented in Figure 11, where a higher FRR indicates better antifouling properties of the membrane. The FRRs of the membranes containing porous TiO_2_ were higher than that of the neat PSf membrane (58%). The 0.5% PTi composite membrane demonstrated an FRR of 80%, while the 1% and 2% PTi composite membranes exhibited almost 96 and 98% FRRs, respectively. This indicates that the addition of PTi enhanced the fouling resistance of the membrane. The trend in the FRR is consistent with the hydrophilicity of the membranes, as indicated in Table 3. Hydrophilic surfaces can absorb water molecules and form a water layer, which inhibits the adsorption of proteins and other fouling agents. Notably, the observed FRR values are higher than those of other carbon nanofillers, such as multiwalled carbon nanotubes [23] and graphene oxide [24], or inorganic nanofillers such as TiO_2_ NPs [36], silica [45], and Fe_3_O_4_ [46]. Furthermore, the incorporation of nanoparticles into the membrane matrix can also improve the uniformity of the membrane’s pore structure, as shown in SEM images, leading to more consistent and predictable flux recovery after cleaning [47]. Nanoparticles can modify the surface chemistry (see Table 2) and morphology of the membrane, creating a more uniform and stable pore structure. These properties can make the membrane less susceptible to damage during cleaning and more likely to recover its initial flux after cleaning.

The water flux and antifouling properties of various composite membranes are compared in Table 4. The use of porous TiO_2_ nanoparticles demonstrated better performance in terms of water flux and antifouling properties. In this study, the modified porous PSf ultrafiltration membranes showed about a 100% increase in water flux. The modified membranes also achieved a 96% flux recovery after the filtration of the bovine serum albumin (BSA) solution, which is among the top 5% of the best-performing membranes listed in Table 4, indicating excellent antifouling performance. 

## 4. Conclusions

To summarize, the incorporation of 0.5 to 2% porous titanium dioxide powder (with a high surface area of 112.84 m^2^/g) into nanocomposite polysulfone ultrafiltration membranes was found to significantly enhance membrane performance, particularly for membrane bioreactor applications. In this study, a hybrid UF/FO process was employed to simulate the osmosis membrane bioreactor process, which has advantages over pressure-driven membrane systems, including lower energy consumption and a reduced fouling tendency. The addition of PTi to the PSF polymer enhanced the membrane’s hydrophilicity and pure water flux while also improving its fouling resistance due to its high hydrophilicity and lower roughness area ratio (r). The optimized membrane, containing 1% PTi, exhibited a water flux of 31.5 L/m^2^h, compared to the neat membrane’s 13.7 L/m^2^h, and a lower solute flux of 0.5 g/m^2^h compared to the other membranes tested. This reduction in reverse solute flux is important since high solute flux can decrease driving forces and, consequently, membrane performance. The MT1 membrane also demonstrated excellent antifouling properties, with a flux recovery ratio of 96%, because it reduced irreversible fouling to less than 5%. These findings highlight the potential of the PTi-infused membrane as a membrane bioreactor (MBR) for wastewater treatment applications.

## Figures and Tables

**Figure 1 membranes-13-00577-f001:**
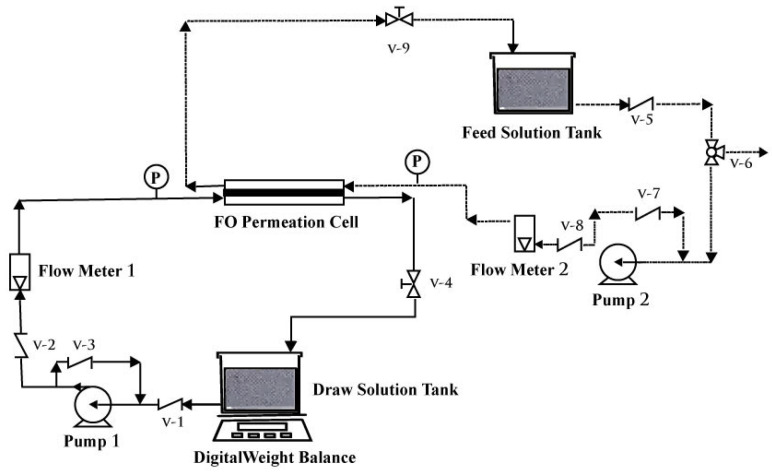
Schematic of a hybrid UF/FO (ultrafiltration/forward osmosis) system used for an osmotic membrane bioreactor (OMBR), where a UF membrane is integrated with the FO process for forward osmosis. Feed solution: BSA as wastewater; draw solution: poly (sodium 4-styrene-sulfonate, 70 kDa); membrane: UF membrane; driving force: osmotic.

**Figure 2 membranes-13-00577-f002:**
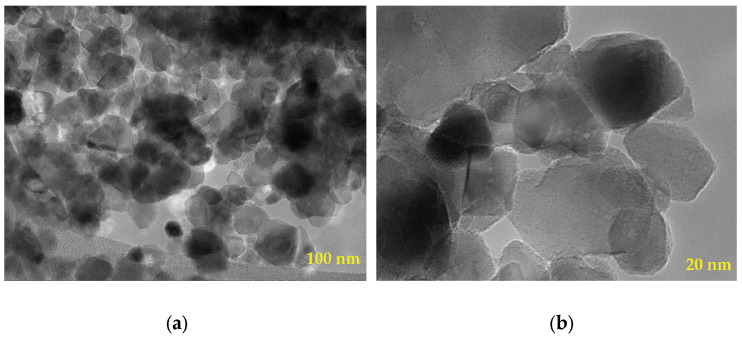
TEM images of titanium oxide nanoparticles: (**a**) 100 nm scale and (**b**) 20 nm scale.

**Figure 3 membranes-13-00577-f003:**
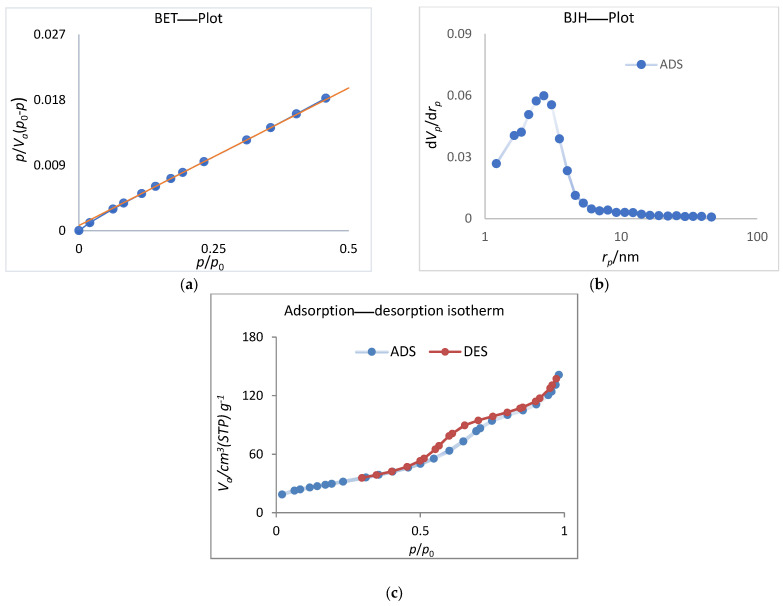
The results of (**a**) BET, (**b**) BJH, and (**c**) nitrogen adsorption–desorption isotherms of the synthesized nanoparticles.

**Figure 4 membranes-13-00577-f004:**
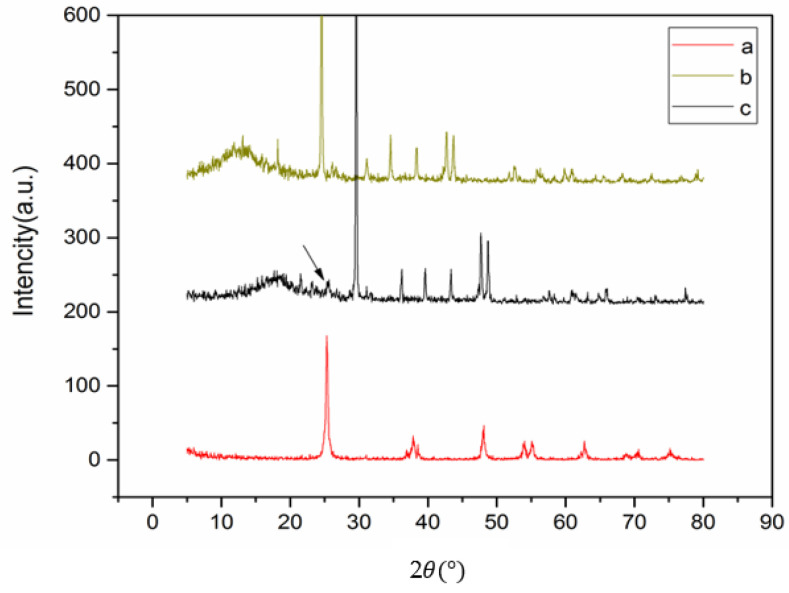
XRD diagram of the crystal structure of PTis (a), nanocomposite membrane (b), and neat membrane (c).

**Figure 5 membranes-13-00577-f005:**
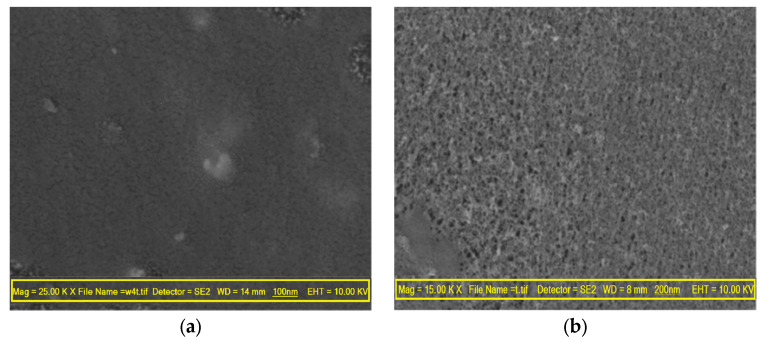
SEM images were taken of the surface structure of the membrane. (**a**) MT0 (control), (**b**) MT0.5, (**c**) MT1, and (**d**) MT2.

**Figure 6 membranes-13-00577-f006:**
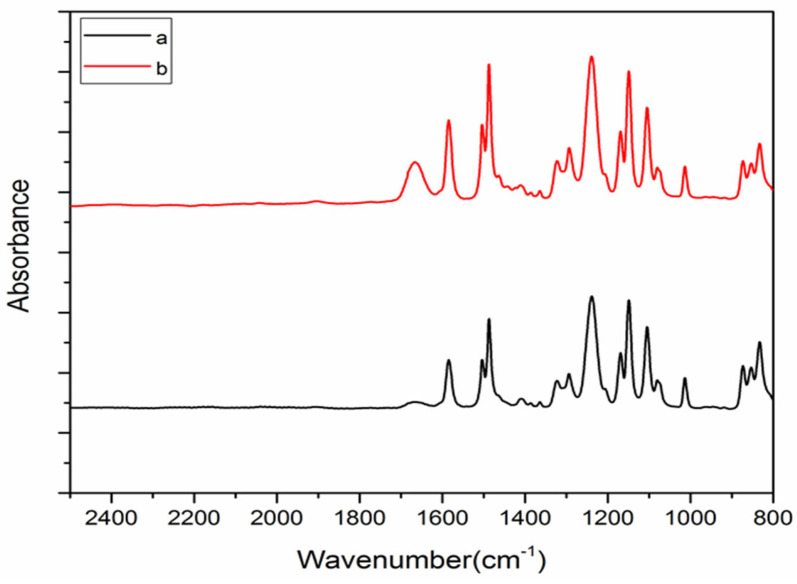
FTIR analysis of (a) PSf membrane and (b) nanocomposite membrane.

**Figure 7 membranes-13-00577-f007:**
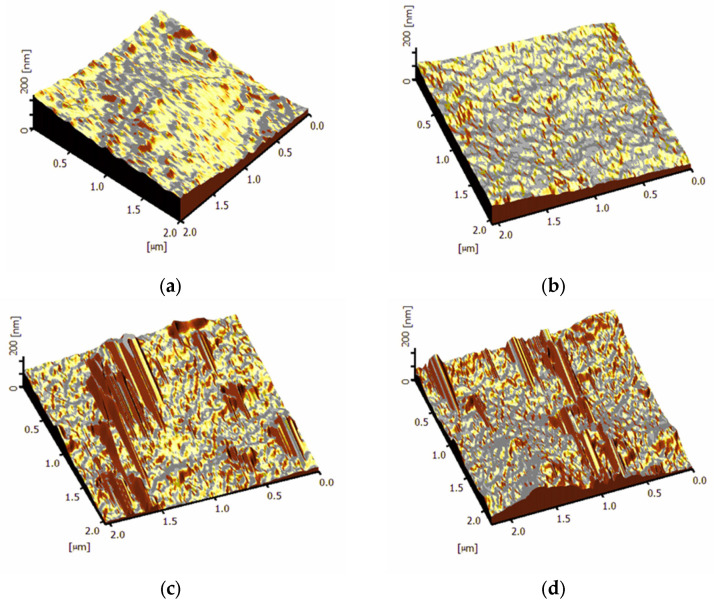
Three-dimensional AFM images of the top surface of PSf membranes prepared from different nanoparticle loadings. (**a**) MT0 (control), (**b**) MT0.5, (**c**) MT1, and (**d**) MT2.

**Figure 8 membranes-13-00577-f008:**
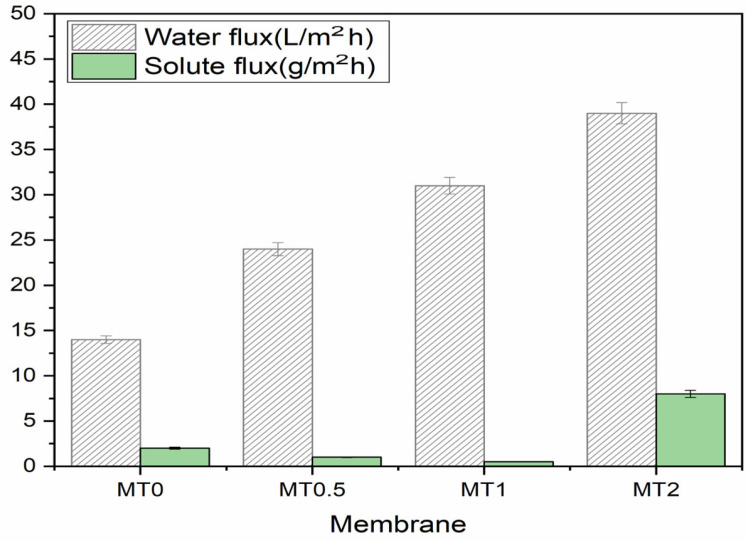
Membrane water and solute flux of PSf and nanocomposite membrane. Test conditions: feed solution: water; draw solution: poly (sodium 4-styrene-sulfonate, 70 kDa); membrane: UF membrane.

**Figure 9 membranes-13-00577-f009:**
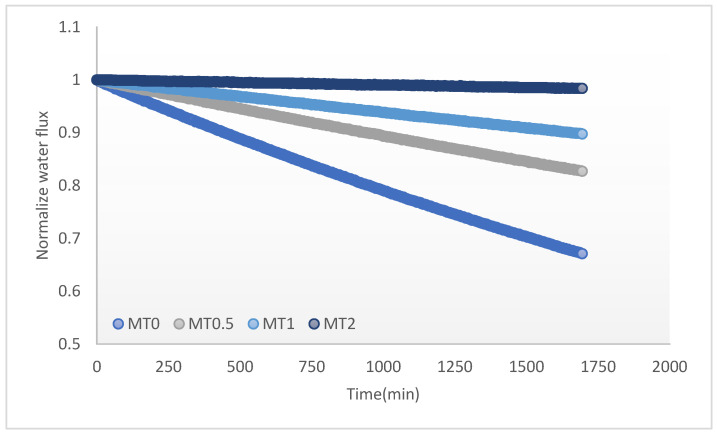
Membrane normalized water flux for composite and nanocomposite membranes. Feed: 400 ppm BSA; draw solute: 0.6 gr of poly (sodium 4-styrene sulfonate); T: 25 °C.

**Figure 10 membranes-13-00577-f010:**
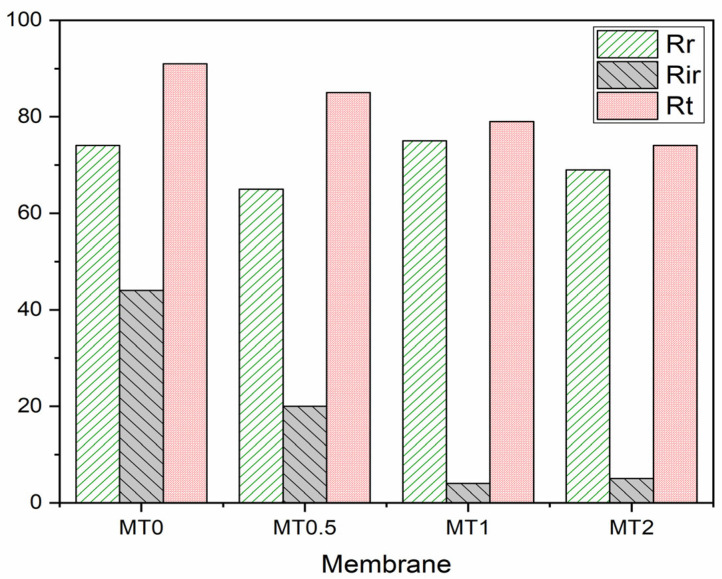
The ratio of deposition resistance of PSf membranes and nanocomposite membranes.

**Figure 11 membranes-13-00577-f011:**
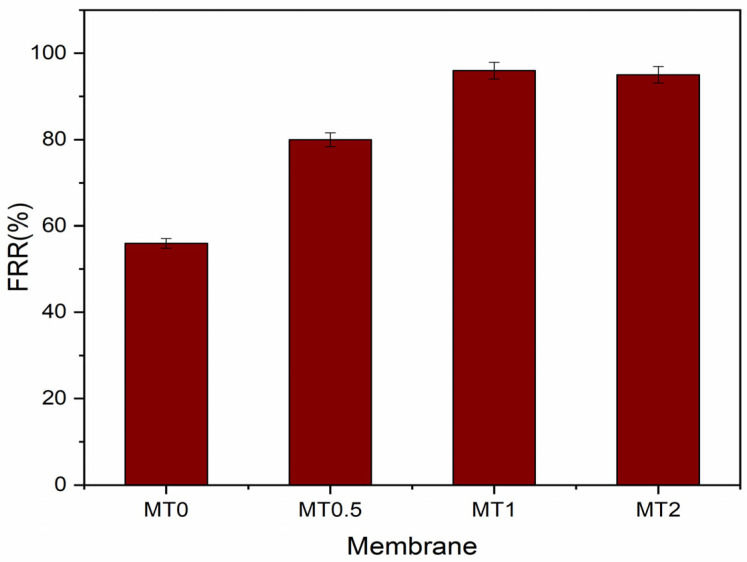
The percentage of water flux return of nanocomposite membranes.

**Table 1 membranes-13-00577-t001:** The composition of embedded membranes with the number of different loadings.

PSF Membrane	PSf (wt%)	PVP (wt%)	NMP (wt%)	PTis
MT0(control)	17.5	0.5	82.0	0.0
MT0.5	17.5	0.5	81.5	0.5
MT1	17.5	0.5	81	1
MT2	17.5	0.5	80.0	2

**Table 2 membranes-13-00577-t002:** AFM analysis, contact angle, surface energy, and *r*.

Membrane	$R_{a\;}$ **(nm)**	$R_{ms}\;$ **(nm)**	$R_{z}$ **(nm)**	**Contact Angle =** θ	$r=\frac{R_{ms}}{R_{a}}$	$\Delta G_{SL}=\gamma_{l}\left(1+\frac{\cos\theta }{r}\right)$
MT0	20.9	27.1	36.49	68	1.29	93.90
MT0.5	28.6	36.2	76.01	64	1.26	98.10
MT1	41.3	50.5	105.7	61	1.22	101.68
MT2	42.6	48.9	122.5	57	1.14	107.54

**Table 3 membranes-13-00577-t003:** Elemental composition (atomic percentage) of the upper surface of PSf and MT membranes by XPS.

Membrane	Carbon (C)	Oxygen (O)	Nitrogen (N)	Titanium (Ti)	Total
MT0	80.46	9.76	9.78	-	100
MT0.5	80.04	10.82	9.07	0.07	100
MT1	78.78	10.30	10.61	0.31	100
MT2	78.80	11.36	9.35	0.49	100

**Table 4 membranes-13-00577-t004:** Water flux and antifouling properties of various composite membranes.

Membrane	Nanoparticle Type	Nanoparticle Concentration	Fouling Resistance	Water Flux L/m2h.bar	Flux Recovery	Particle Size	References
PSf	MT1	1%	High	52.5 *	96%	7–40 nm	This work
PSf	TNT	0.1%	Low	58	78%	20–40 nm	[30,48]
Cellulose acetate	SiO_2_	10%	Moderate	100	90%	250 nm	[49]
PES	TiO_2_-ZnO	0.5%	Low	10.6	76%	-	[50]
PVDF	Al_2_O_3_	2%	High	150	96%	10 nm	[51]
PVDF	CuO	1%	Moderate	480	90%	-	[52]
PVDF	Fe_3_O_4_@Xanthan gum	0.2	Low	50	66.5%	10–40	[53]
PES	GO	3%	Moderate	270	92%	-	[54]

* 31.5 L/m^2^h at 0.6 bar.

## Data Availability

We can make the data available upon request.

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
