# Peer review of "Integration of Porous Nanomaterial-Infused Membrane in UF/FO Membrane Hybrid for Simulated Osmosis Membrane Bioreactor (OsMBR) Process"

_membranes, 2023, doi:10.3390/membranes13060577_

Round 1

Reviewer 1 Report (Previous Reviewer 1)

The comments have been adressed well in the revised manuscript to improve its quality.

Author Response

Thank you very much for accepting our revision.

Reviewer 2 Report (Previous Reviewer 2)

The work was improved, but questions and comments are still left. Which material does figure 3 refer to? Is it titanium oxide particles or the membrane with particles? Figure 4 is incorrect. It is standard to write 2 Theta, not 2 Tata. What is the value of gamma_l in table 3? The comparative analysis should be determined by the task at hand - flux, retention, etc. A qualitative assessment of "Fouling Resistance" cannot give an adequate understanding of the membrane. Also the advantages of these membranes over other membranes should be highlighted.

Author Response

The work was improved, but questions and comments are still left.

Thank you for accepting our revision. We have revised the manuscript once again based on your new comments.

Which material does Figure 3 refer to? Is it titanium oxide particles or the membrane with particles?

The material was porous TiO2. It is highlighted on line 241.  

Figure 4 is incorrect. It is standard to write 2 Theta, not 2 Tata.

Thank you so much for your precise feedback about the manuscript. We have revised Figure 4 accordingly.

What is the value of gamma l in Table 3?

We have added more explanation about the formula and included it as a new formula, Formula 7 on lines 203 to 224.

The comparative analysis should be determined by the task - flux, retention, etc.

We have attempted to revise Table 5 and its corresponding explanation from lines 468 to 478.

 A qualitative assessment of "Fouling Resistance" cannot adequately understand the membrane.

Fouling is a common problem in membrane filtration processes, where contaminants build up on the membrane surface and reduce its performance over time. Fouling resistance is a measure of a membrane's ability to resist fouling, and it is an essential parameter for assessing the effectiveness and durability of membrane systems. A qualitative assessment of fouling resistance can provide valuable insights into the membrane's performance and suitability for a given application. Also, a qualitative assessment of fouling resistance can provide valuable insights into the membrane's performance and suitability for a given application. By monitoring the appearance of the membrane, flux decline, fouling index, and cleaning effectiveness, the operator can optimize the membrane's performance, extend its lifespan, and reduce operational costs.

Also, the advantages of these membranes over other membranes should be highlighted.

We have attempted to revise the conclusion accordingly.

Reviewer 3 Report (Previous Reviewer 3)

The authors gave good feedback on my earlier comments. I think I have no more questions.

Author Response

Thank you very much for accepting our revision.

Reviewer 4 Report (Previous Reviewer 4)

The authors have refined the manuscript.

Some spelling mistakes found in the manuscript.

Author Response

Thank you for bringing up your concern. We have thoroughly edited and polished the manuscript.

Reviewer 5 Report (Previous Reviewer 5)

The authors performed some modifications on their manuscript but these are not sufficient to modify my opinion on the paper, especially because none of my initial comment is answered in the modified version. For my point of view, this manuscript should be rejected due to the amount of modifications needed.

The language editing of the manuscript remains insufficient and didn't concern the whole manuscript, which represents a major issue.

Author Response

Thank you for bringing up your concern. We have thoroughly edited and polished the manuscript.

Round 2

Reviewer 2 Report (Previous Reviewer 2)

Unfortunately, the authors still do not understand the terminology for determining the parameters of porous materials by low-temperature nitrogen sorption. First, an isotherm is obtained and then both the specific surface area by BET and the pore size distribution function by BJH are determined.

In the caption of figure 3, the membrane material is indicated, but in the text the titanium oxide particle is mentioned. There is also a question as to why there is no comparison between the isotherms of the particles and the material.

Figure 3 (b) shows the wrong dimensions for the pore size distribution function.

Adding particles to the polymer matrix is common practice for obtaining new or improving existing material properties. The results of other work presented in Table 5 show that the proposed material does not have characteristics superior to those previously obtained.

Author Response

Unfortunately, the authors still do not understand the terminology for determining the parameters of porous materials by low-temperature nitrogen sorption. First, an isotherm is obtained and then both the specific surface area by BET and the pore size distribution function by BJH are determined.

Previously, we held the assumption that the methodology for BET analysis may not be crucial and that only the interpretation of the results was significant. However, we have tried to incorporate this information into our manuscript. We added more information related to BET and BJH methods in the manuscript on lines 239 to 252.

In the caption of Figure 3, the membrane material is indicated, but in the text, the titanium oxide particle is mentioned. There is also a question as to why there is no comparison between the isotherms of the particles and the material.

Thank you for bringing this to our attention. We only utilized BET analysis for the nanoparticles, so we have updated the caption accordingly.

We are unsure of the necessity of comparing the isotherms of the particles and the material, as our analysis was solely focused on examining the porosity and surface area properties of our nano-sized particles.

Figure 3 (b) shows the wrong dimensions for the pore size distribution function.

The x-axis of the figure shows the pore size radius, which ranges from 20 to 75 A. The original range of 20 to 75 Angstroms (A) should be divided by 10 to report the pore size in nanometers.

Adding particles to the polymer matrix is common practice for obtaining new or improving existing material properties. The results of other work presented in Table 5 show that the proposed material does not have characteristics superior to those previously obtained.

One advantage of this work is that, while the water flux is comparable to or even higher than that reported in Table 5, it was tested in an FO system that relies on osmotic pressure, resulting in lower energy consumption. In contrast, the other works reported in Table 5 are based on pressure-driven systems. Additionally, the membrane flux recovery of this work is among the highest reported.

Reviewer 5 Report (Previous Reviewer 5)

The authors performed some modifications on the major drawbacks of their manuscript which is now suitable for publication, even if they could have improved further their manuscript.

Some basic revisions have been made. English is now understandable even if a global polishing would be positive.

Author Response

Thank you for accepting our revision. 

Round 3

Reviewer 2 Report (Previous Reviewer 2)

The description of the BET and BJH methods is not necessary in so much detail. It would be sufficient to write that the specific surface area (BET), the specific pore volume and the pore size distribution function (BJH) have been calculated from the low-temperature nitrogen adsorption-desorption isotherm (Fig. 3a). The dimensionality of the pore size distribution function is [cc/g/A] or [cc/g/nm]. Also the dimensionality of the amount of adsorbed gas is usually written as [cc/g] under standard conditions (STP).

Author Response

Thank you for the comment. We revised this section properly.

This manuscript is a resubmission of an earlier submission. The following is a list of the peer review reports and author responses from that submission.

Round 1

Reviewer 1 Report

The article presents the fabrication and application of porous nanomaterial infused membranes for osmotic membrane bioreactor for wastewater treatment. It is a relevant study, but with some conclusions that are not supported by the data, and the authors do not succeed to describe how the membrane can work as an osmosis membrane.

My concerns are:

1. The authors write that the membranes are ultrafiltration nanocomposite systems, but don't explain how these can permeate water while rejecting low Mw compounds, e.g. salts. It is a porous membrane, with a reported mean pore radius of 40 Å, which is way to large to retain salts. 

2. It is written that the membrane is used for OsMBR, but no information about the system and the operating parameters have been given.

3. there are no details about the operational conditions of the osmotic membranes. What was the osmotic gradient? Was there any hydraulic pressure gradient contributing to high flux (and thereby high salt reverse flux), e.g. by level differences?

4. Eq. 3 doesn't make sense to describe rejection, as it doesn't take into account the concenctration effect on the feed side.

5. it is not argued in the introduction, why the incorporation of TiO2 nanoparticles in membranes may be beneficial for FO membrane development. 

Reviewer 2 Report

The topic of the study is interesting and up to date. Unfortunately, the authors have not succeeded in advancing this line of the research. Not achieved the results presented in [Bae, Tae-Hyun, and Tae-Moon Tak. "Effect of TiO2 nanoparticles on fouling mitigation of ultrafiltration membranes for activated sludge filtration." Journal of Membrane Science 249.1-2 (2005): 1-8]. The article needs a serious revision. There are also erroneous sentences (387-388, The FRRs of the membranes containing nanotubes were higher than that of the neat PSf membrane (58%)) and misunderstanding of the authors of the research methods (207, In Figure 3, the results of the Brunauer-Emmett-Teller (BET)...).

Reviewer 3 Report

This manuscript entitled “Improving Wastewater Treatment Efficiency with a Porous Nanomaterial-Infused Membrane in the Osmosis Membrane Bioreactor (OsMBR) Process”, by Ahmadreaz Zahedipoor et al., presents the performance of porous titanium dioxide incorporated nanocomposite membranes and its antifouling properties in OsMBR process. It’s an interesting topic. However, the correction and improvement are needed.

Q1 The author mentioned that lots of nanomaterials were used for fabricating the nanocomposite membranes (line 52-61). I think the author should make it clearer why choose the PTi for fabricating the high performance antifouling membranes. What unique advantage does PTi have over other nanomaterials.

Q2 The type of draw solution and its concentration should be specified in the test. In fact, I think this is a forward osmosis (FO) process, not an OsMBR process.

Q3 In Figure 5, are the four SEM images in the same magnification? Because the MT2 membrane exhibited an abnormal dense morphology than others in Figure 5d.

I also suggest adding the cross-sectional morphologies of the four membranes to observe the effect of PTi on membrane pore structure.

Q4 The author mentioned “such as C=O (carbonyl) at around 1730 cm-1, O-H (hydroxyl) at around 3400 cm-1, and C-H (alkyl) at around 2900 cm-1” in line 253-255. However, in Figure 6, the X-axis ranged from 800 to 2500 cm-1.

Q5 In the description of Figure 7, the author claimed two contradictory conclusions that both the unmodified membrane (line 271-273) and the modified membrane (line 284-286) had anti-fouling properties. Please supply the relevant explanation.

Reviewer 4 Report

1. Please further elaborate the past findings of membrane modified with nanoparticles and applied in OsMBRs especially the challenges still pending for resolve.

2. The size of TiO2 nanoparticles viewed under TEM must be supported with appropriate measurement technique/software. 

3. The authors claimed that the pore size of membrane decreased with increase in nanoparticle concentration. Please further elaborate as compared to control membrane MT0, the latter has less pore (based on Figure 5). Furthermore, Figure 5(d) looks totally different from the rest. Any explanation for this? Details of analysis should be included for Figure 5.

4. Line 269-272. The two sentences here have contradictory claims. Please verify.

5. Line 277-278. Please explain further.

6. Line 316-319. Further clarification is required. What is lower return salt? And how the limited compatibility be linked to the performance?

7. Why salt flux increased substantially for MT2?

8. Line 339-343. Please explain based on the data shown in Figure 9. 

9. Where is the evidence for claim made in Line 343-345?

10. Caption of Figure 9 - please include testing conditions

11. Please include a section to compare the characteristics and performance of the modified membrane in this study with other reported studies. 

Reviewer 5 Report

This manuscript by Zahedipoor et al. aims to assess the impact of NPTis addition on membrane properties.

The scope of this manuscript is of interest for the readership of Membranes, unfortunately some basic requirements are not respected in this work. (i) the context was not correctly drawn and didn't correspond to the performed experiments, (ii) the performed experiments are not correctly explained and it is impossible to understand what was precisely realized as plenty of details still lacking, (iii) the figures displayed curves and points that are not exploitable as the precise conditions are not present, (iv) the discussion on the figures are often disconnected from the obtained data (e.g. FTIR and XRD), (v) you repeat several times throughout the manuscript some sentences as "the presence of NPs alter the surface ... creating a more hydrophilic and rougher surface" this is not correct, (vi) you speak about salt but it is not clear what kind of salt you consider, (vii) the choice of 1% instead of 2% is not clear as well and it is a pity as it is the only major result of your work, and (viii) as for the introduction, the conclusion is not informative on what you've precisely done, finally the title of the manuscript didn't reflect its content.

Due to these major limitations, I recommend to reject this manuscript, which is below the standards of Membranes.